# Evolution of Physical Training in Police Academies: Comparing Fitness Variables

**DOI:** 10.3390/healthcare11020261

**Published:** 2023-01-14

**Authors:** Bridget Melton, Gregory Ryan, Victor Zuege, Haresh Rochani, Derick Anglin, Joseph Dulla

**Affiliations:** 1Department of Health Sciences and Kinesiology, Water College of Health, Georgia Southern University, Statesboro, GA 30460, USA; 2Exercise and Sport Science, College of nursing and Health Sciences, Piedmont University, Demorest, GA 30535, USA; 3Federal Law Enforcement Training Center, Brunswick, GA 31524, USA; 4Department of Biostatistics, Jing-Ping Hsu College of Public Health, Georgia Southern University, Statesboro, GA 30460, USA

**Keywords:** tactical athlete, law enforcement, fitness trends, police academies

## Abstract

The purpose of this study was to evaluate the effectiveness of three different physical training approaches to improving cadets’ fitness variables. Retrospective data for male and female land management law enforcement officers attending a 15-week training program at three separate time points were provided for analysis. The time points reflected the three different training approaches, including calisthenic training (CT) = 83, functional fitness training (FT) = 90, and strength training (ST) = 110. Inferential data analysis was used to find which mode of exercise had the greatest impact on body composition, cardiovascular endurance, muscular strength, agility, and flexibility. All groups displayed decreases in body fat percentage, with weight loss being more significant within the CT and FT groups, while the ST group increased in body weight. The CT group had the greatest flexibility increases compared to the FT and ST groups. ST training elicited significantly smaller changes in cardiovascular endurance than the FT and CT groups. ST training showed greater improvements in lean mass, while CT and FT showed greater increases in flexibility and endurance. These results suggest that training protocols can increase performance and optimize the abilities to perform job tasks in tactical athletes.

## 1. Introduction

The diversity of occupational tasks of law enforcement officers (LEO) ranges from sedentary, with long periods of inactivity, to highly physically demanding and potentially life-threatening activities. It has been estimated that 80–90% of a police officer’s workday is composed of sedentary behaviors, such as sitting, standing, and slow walking [1]. In a study by Ramsey, the rates of hypertension and being overweight were greater within the police department compared with the general population [2]. However, with LEOs having to respond to a situation, they must have an occupational fitness level to perform unpredictable and stressful bursts of intense physical activity. LEOs are at risk for potential dangers daily, which can be physically and mentally demanding. Therefore, it is vital that LEOs increase and maintain durability to perform duties to ensure longevity and service to our society [3]. Several law enforcement duties (e.g., apprehending a subject, empty-hand control techniques, forcing entry) require adequate cardiorespiratory and muscular fitness [4,5]. Furthermore, LEOs who conduct routine uniformed patrol enforcement often carry external loads, including 7–10 kg vests, and Special Weapons and Tactics (SWAT) units can carry up to 23 kg of gear, including weapon systems, body armor, and communication systems [1,6]. These occupational requirements and activities put LEOs at risk of both cardiovascular events [7] and musculoskeletal injuries [8]. With LEOs having more physical demands placed on the body, it is important that these officers have good physical parameters, including mobility, muscular strength and endurance, and cardiorespiratory fitness [1].

Law enforcement agencies commonly use internal or external police academies as a means of training recruits for a career in law enforcement. Training programs are equipped to help police officers develop multiple skills and qualities to adapt to real-world situations, mentally and physically. These skills can include communication, comprehension of policies and laws, decision-making, and problem-solving skills [9]. Academies also teach the necessary skills and procedures (e.g., carrying, dragging, jumping, etc.) for working as an LEO and prepare recruits for the physical and psychological challenges of working in law enforcement [10,11]. Physical abilities training (including general physical/lifetime fitness) is a fundamental part of police academies and sets the precedence for years of law enforcement service. The physical training approaches in police academies have varied over the years [2,12].

As exercise trends have evolved over the years, so have the physical training practices used in police academies. Traditionally, police academies utilized calisthenic/aerobic conditioning (CT) (e.g., formation runs) and muscular endurance (e.g., bodyweight exercises), as seen in the military boot camps in the late 1990s. However, with the high injury rates among recruits [13], military branches have re-evaluated training programs with a focus on occupational task needs assessments and shifted from traditional push-ups and sit-ups to physical training practices that are more specific to the performance of routine and common daily tasks such as lifting and carrying [14]. Similar injury types and rates were also observed in police academies, especially among low-conditioned individuals [15]. In the early 2000s, some police academies shifted to a high-intensity interval training (HIIT) or functional training (FT) approach, resembling the popular type of workout of the time [16]. Recently, strength training (ST) programs with planned periodization have been incorporated into police academies [17]. Muscular strength and power are necessary to accomplish certain job tasks, such as physically taking a non-compliant individual into custody or physically subduing an arrestee, and these concepts are highly linked with an individual’s overall strength [18]. Additionally, numerous large-scale studies of the physical job demands of LEOs assigned to patrol functions have consistently ranked muscular power, anaerobic capacity, and strength as contributing to critical tasks more than muscular endurance or aerobic capacity [19]. Furthermore, due to the lack of fitness, particularly muscular strength, the chances of the LEO being able to perform the task properly are reduced, which may contribute to higher levels of force in critical situations that could lead to elevated risks of injury [20]. Lastly, given the largely sedentary nature of police activity, focusing on calisthenics or aerobic training may not be as job-specific as initially theorized. Therefore, increasing the intensity and volume of calisthenics and aerobic training in a sedentary population can lead to increased injury risk [21]. The purpose of this study was to retrospectively compare the impact of three different exercise focuses on police academy cadet fitness testing variables.

## 2. Materials and Methods

### 2.1. Research Design and Participant Selection

Retrospective data for male and female land management law enforcement officers attending a 15-week training program at a United States Federal Law Enforcement Training Center at three separate time points were provided for analysis.

The training center was located in the southeastern portion of the United States and served more than 120 law enforcement agencies. The training complex is encompassed over three acres, including mat rooms, classrooms, weight rooms, a gymnasium, and other facilities devoted to training and fitness. The Land Management Police Training Program (LMPT) was selected in consultation with the commanding officers, as this group depicted an average intensity of physical training. The LMPT is designed to meet the entry-level training needs of federal law enforcement officers responsible for protecting natural resources and public lands. A total of 283 fitness assessments were randomly sampled to represent the different fitness approaches: CT = 83 (69 male, 14 female); FT = 90 (79 male, 11 female); and ST = 110 (96 male, 14 female) at 3 points in time as noted below.

### 2.2. Training Programs

Over the decades, the sampled police academy training center modified its physical training approach to be in line with the current exercise trends at the given time. There have been three distinct trends employed at the training center, CT from 1990 to 2000, FT from the early 2000s to 2008, and ST from 2009 to the present. Table 1 depicts additional details, such as the frequency of sessions per week and sample daily programs for the 3 listed trends/approaches.

### 2.3. Measures

Upon arrival at the academy, recruits were tested during weeks 1 (pre), week 7 (mid), and week 14 (post). Testing was completed during a regularly scheduled physical training session at 0700 h (7:00 a.m. local time). The testing sequence included body fat, flexibility, muscular strength, agility, and endurance; testing sessions lasted for approximately two hours. The testing order consistently had body composition first and the 1.5 mile (2.4 km) run last; however, there was some variation per session on the flexibility, strength, and agility due to logistics and time management.

#### 2.3.1. Body Fat

Recruit body fat was measured using a 3-site skinfold test with a trained test administrator. Male recruits were measured at the chest, abdomen, and upper thigh. Female recruits were measured at the triceps, iliac crest (hip), and upper thigh in accordance with the traditional Jackson-Pollock methodology.

#### 2.3.2. Flexibility

The flexibility of the lower back and hamstrings was measured using a standard sit-and-reach box. Participants sat on the ground with their shoes off and back flat against a wall and were instructed to bend at the hips, pushing the measurement gauge as far forward as possible in a slow and controlled movement. Test administrators ensured that participants maintained extended legs throughout each trial. Three trials were conducted, and the highest value in inches was incorporated into the results of this study.

#### 2.3.3. Muscular Strength

Upper body muscular strength was evaluated utilizing the bench press with trained exercise professionals following National Strength and Conditioning Association guidelines [1]. The maximum weight lifted and relative weight (maximal weight divided by body weight) were recorded. 

#### 2.3.4. Agility

Agility was tested using the Illinois Agility Test. The path length was 10 m long by 5 m wide (the distance between the start and finish). The start, finish, and 2 turning points were indicated by 4 cones (Figure 1). Four more cones were placed equidistantly in the middle of the course. The central cones were placed 3.33 m from each other. The recruit began to run when triggered by the signal, as indicated in the diagram. The time to complete the course was measured in seconds. The measurement of time in the Illinois Agility Test was performed by stopwatch with an accuracy of 0.01 s [22].

#### 2.3.5. Cardiovascular Endurance

The 1.5 mile (2.4 km) run was used to assess cardiovascular endurance. The test was performed on a standard 440 yard (400 m) road-surfaced outdoor track, with the test requiring approximately 10–15 min. The time to complete the 1.5 mile run was recorded to the nearest second with a digital stopwatch [22].

### 2.4. Statistical Approach

Descriptive statistics and frequency analyses were conducted for all variables to understand and describe the distribution of the sample collected over time. Paired tests were performed to compare the pre and post scores for performance measures. The main goal of the study was to evaluate the effectiveness of physical training approaches on performance measures over time. Generalized linear models are widely used in healthcare to identify changes over time and provide an overview of cause-and-effect relationships [23]. Moreover, physical ability and performance measures are presented in delta change (Δ%) which is the average change from the pre/post change of each of the participants. Significance was set at *p* ≤ 0.05, and all data are presented as means ± SD. The general linear model analysis is well-adapted and tested in SAS. Therefore, we used the latest SAS statistical software (version 9.4) to conduct all our analyses.

## 3. Results

There were 100 recruits measured in 2001, 96 in 2011, and 115 in 2018; gender percentages for each period are shown in Table 2 (Gender Distribution by Time). Across the three recruit classes (10 in 2001, five in 2011, and five in the 2018 sample), there was a total of 20 subjects that may not have completed post-measurement testing or were not available for the testing, see Table 2.

Table 3 represents the summary statistics for each type of training approach (year cohort); significant differences were found from pre to post as indicated with (*p*) and the corresponding physical ability and performance measures presented in delta change (Δ%) which was the average change from the pre/post change of each of the 100 participants.

Table 4 shows the effect of independent variables (age, gender and time) on performance measures. There was a significant effect of gender on the weight of the recruit in the premeasurement; females were 32.8 pounds lighter than males (*p*-value < 0.01). Compared to the year 2001, the Pre weight of the recruits from the year 2011 was 9.39 pounds higher (*p*-value = 0.03).

Table 4 shows that females had significantly higher flexibility than males (*p*-value < 0.01); more specifically, the trunk flexibility was 2.34 inches higher in females.

Table 4 shows a significant effect of gender, age, and time on the bench press in the premeasurement as well. Females could bench press 98.73 pounds less compared to males (*p*-value < 0.01). Furthermore, as age increased by 1 year, on average, the bench press decreased by 1.48 pounds (*p*-value < 0.01). Lastly, recruits in the year 2018 can bench press a weight 18.74 pounds higher than in the year 2001 (*p*-value = 0.0003).

Table 4 shows that there was a significant effect of gender on the premeasurement of the percentage of weight the recruit could bench press relative to their own weight (*p*-value < 0.01); the percentage of weight females could bench press was 40.89% lower compared to males. Moreover, each increase in age by 1 year corresponds to a lowering of 0.96% (*p*-value < 0.001). Lastly, in 2018, the premeasurement of the weight the recruits could bench press was 6.61% higher (*p*-value = 0.02).

Females had, on average, a run time 2.73 s higher compared to males for the premeasurement (*p*-value < 0.01). There was a significant effect of age; with an increase in age of 1 year, run time increased by 0.08 s (*p*-value < 0.01). Compared to 2001, the run time in 2018 was, on average, 0.56 s lower (*p*-value = 0.02).

Table 4 demonstrates that agility was, on average, greater in females by 2.33 s (*p*-value < 0.01). Age had a significant effect, where it can be seen that an increase of 1 year in age increased agility by 0.07 s (*p*-value < 0.01). Lastly, there was a significant effect of time, where it can be observed that, in 2011, agility was 0.28 s higher than in the year 2001 (*p*-value = 0.03).

When considering the premeasurement of body fat percentage, it could be observed that it was, on average, 8.99% higher in females compared to males (*p*-value < 0.01). Age also showed a significant effect, where an increase of 1 year showed an increase in body fat by 0.21% (*p*-value = 0.01). Lastly, body fat was 1.83% lower in 2018 compared to 2001 (*p*-value = 0.01).

## 4. Discussion

The primary finding of this study was that changes in fitness trends in the sampled police academy were observed and varied across fitness programs (points in time). Differences were seen in body fat, trunk flexibility, cardiovascular endurance, and strength. No significant changes were observed in body weight or agility. This study was unique in that it looked at the influence of changes in recruits’ fitness levels coming into law enforcement academy, and it also looked at the effects of different fitness trends on fitness changes over a 12-week academy.

### 4.1. Body Composition

Over the last 20 years, the average weight of LEO academy recruits at this training location has increased by approximately 5 kg, and the majority of LEOs categorized as overweight [24]. These trends are consistent with overall weight gains in the United States [24], further highlighting the need to address body composition in LEO academy programs. The current study found that weight loss was achieved following completion of CT (~5lb) and FT (~2lb) programs, while an ST-focused program resulted in a ~2 lbs increase in body weight which is consistent with previous research [25,26]. Weight gain is common in ST programs, due to potential increased muscle mass, with BF% maintaining or reducing. A ~3–4% reduction in total BF% (reflecting a 15% reduction on average) was found across all groups following the completion of training. As such, ST was the only training modality that noted an increase in BW and a reduction of BF%, indicating a potential gain of fat-free or muscle mass. This may suggest that ST is an appropriate training modality for cadets who either need to gain or maintain body weight or gain fat-free mass, while FT- or CT-focused training would be beneficial for body weight and body fat reduction [27].

### 4.2. Flexibility

This study did not find any significant changes in flexibility among the different modes, with an average of a 5% increase. However, only FT and ST did have significant changes, whereas no change was detected in CT. The current investigation results suggest that a fitness program focused primarily on CT lacks the appropriate stimulus to improve flexibility to the same degree as other programs in this population. Numerous benefits are associated with flexibility programming, including increased range of motion, stabilization of the joints, and feelings of relaxation. Flexibility training is commonly used to improve performance, specifically muscular strength [28]. Appropriate flexibility is also critical for optimal LEO performance across physical tasks. Targeting the flexibility fitness variability for performance may be beneficial along with instruction in resilience and stress management techniques [29]; however, that was out of the scope of this study but is recommended in future research.

### 4.3. Performance Variables

This study discovered a trend of incoming recruits having a higher level of upper body strength; in 2018, recruits pressed an average of 105% of their weight, compared with only 99% and 97% in 2001 and 2008, respectively. Interestingly, when strength was measured by the bench press, there were no significant differences in upper body strength gains between training styles; all groups, on average, gained a 10-pounds of upper body strength and an average of 10% increase for bench press relative to body weight. Nevertheless, looking at the programs individually, FT and ST both had significant changes in pre- and post-bench. Previous research is in line with these findings of maintenance and slight increases in upper body strength with specialized training units [30]. This could be explained in part by the training load. Total Training Load may have been reduced in with ST. The frequency and duration of training went from 3–5 days per week, 1.5 h for CT and 4.5 h for FT per week, to 3 days per with about 0.5 h plus defensive tactic time or a total of 2.5 plus hours for ST. Defense tactics can combine anaerobic and aerobic energy systems and rely on muscle power. The actual intensity level set and completed for each training mode was not recorded; however, it is well established the training loads must be at a specific level to produce training effects [1].

Like upper body strength, no statistically significant difference was observed between training styles for the three groups for agility, with all training style modalities resulting in improved Illinois Agility time to completion. Again, the results did show a significant change in the FT and ST pre and post changes. Both FT and ST have more power movement compared to CT, which is more aerobic and muscle endurance-focused. Agility is a critical skill in law enforcement, and lack of agility could be a source of on-the-job injuries; the individual needs to be proficient with changing directions, acceleration, and deceleration [3,31]. If recruits are deficient or struggle with agility, as indicated by slower scores or visual observations in this performance measure, recruits should be trained to become proficient. It appears that any training modality focus is sufficient in improving agility in this population. This may be due to the non-specific training each of these programs had for agility, including anaerobic and muscle fitness benefits.

Furthermore, upper body strength is essential in LEOs, and ST should be incorporated into training programs. Including ST in programs will build resiliency along with increased performance in case of the need to perform emergency physical tasks in the workforce [32].

While the FT and ST training modalities resulted in similar significant changes in strength and agility performance, a primary focus on ST resulted in smaller improvements in aerobic performance in our findings. Agencies should consider this distribution of training effect as they review unique and specific agency law enforcement officer physical task distributions. For example, an agency like the United States Customs and Border Protection may benefit from an equal distribution of physical abilities to support long walking distances with external loads in harsh environments, whereas an agency with primarily metropolitan or suburban patrol responsibilities may benefit more from an increased focus on strength, power, and anaerobic capacity. Again, specific physical performance job tasks are unique to agencies and should be a primary area of focus for training. Further, all programs successfully improved the 1.5 miles (2.4 km) run time, but ST resulted in approximately half as much improvement compared to CT and FT. Depending on the documented physical job demands of the specific LEO’s role, the observed improvements and training modalities may or may not be as balanced as desired. For example, some assignments, such as general uniformed patrol, require greater emphasis on stability, strength, power, anaerobic capacity, and agility versus endurance and aerobic capacity [19].

In the tested population, it is possible that recruits could benefit from an additional training stimulus or program adjustments to further develop aerobic capacity depending on documented physical job demands. Tactical strength coaches should look to incorporate a training program that helps recruits maximize their abilities on all tested parameters and documented job demands without overly focusing on gains in one area unless dictated by the physical job demands of the roles trainees plan to occupy.

### 4.4. Limitation

There were several limitations with that that need to be acknowledged. First is the random selection of recruit classes close to 10 years apart within each recruit class and training protocol. Randomly selecting different recruit classes might not fully represent the observed results; additionally, the cultural tone surrounding exercise between the data points may have influenced the recruits’ effort levels. Second, data on the racial and cultural backgrounds of the recruits were not available for this study, so insight into how these demographics may influence physical performance can not be discussed. Additionally, the focus of this study was limited to performance on measures of physical fitness only and not the performance of job tasks as many additional variables such as weather, fatigue, shift length, number of officers present, and others’ impact performance on job tasks. It is recommended to explore these racial and cultural backgrounds relative to performance on physical fitness assessments in future research. Third, testing protocols were limited. Upper body muscular strength was measured, but maximal muscular strength for the lower body was not assessed. While upper body muscular strength is critical, lower body strength is also essential to successfully completing required occupational tasks. Although form and technique could have been an issue when assessing lower body strength, submaximal values could have been analyzed to observe lower body strength within recruits. Fourth, aerobic capacity was measured within the recruits, but anaerobic capacity was not assessed. As LE occupational tasks require explosive movements and sprints, using job-relevant assessments such as the 20 m Multistage Fitness Test (20M MSFT) or others could increase the robustness of the analysis. Finally, while statistical significance was observed, it is worth noting the wide variation in the data, which may be due to several factors, including the range of fitness levels of recruits who enter this program and the limited sample sizes of this data set.

## 5. Conclusions

Despite the emphasis on the physical training mode used in police academies, this study found each mode positively impacted fitness variables. Differences in body fat, trunk flexibility, and cardiovascular endurance were observed based on training mode focus. Additionally, FT and ST had more significant positive changes in their individual pre and post scores for performance and fitness variables than CT. As discrete physical job tasks, criticality, frequency, and duration can vary greatly between and within agencies (patrol vs. administration vs. investigator vs. specialist/SWAT roles) along with load carriage requirements, it is recommended to review and consider the specific physical job demand profile when designing the recruit physical training program to optimize the fitness outcomes of recruit officers based on roles they are expected to fill upon academy graduation. For example, job demand profiles that require greater strength, power, and anaerobic capacity (most metropolitan, urban, and suburban uniformed “traditional patrol-type” duties) could benefit from the emphasis on FT or ST, where job demand profiles that require greater emphasis on endurance and aerobic capacity might benefit more from an increased focus on CT programming. Future research should also examine the effects of similar programming (CT, FT, ST) with larger and more varied sample populations and sample sizes and different populations both at this training location and other similarly matched LE recruit training facilities and agencies. Examining injury rates relative to training modality (CT, FT, and ST) could also be beneficial. Tactical strength coaches should look to incorporate a training program that helps recruits maximize their abilities on all tested parameters and documented job demands without overly focusing on gains in one area unless dictated by the physical job demands of the roles trainees plan to occupy. Further, instilling lifelong skills and passion for physical activity and exercise in recruits may mitigate chronic disease and strengthen positive mental health coping mechanisms.

## Figures and Tables

**Figure 1 healthcare-11-00261-f001:**
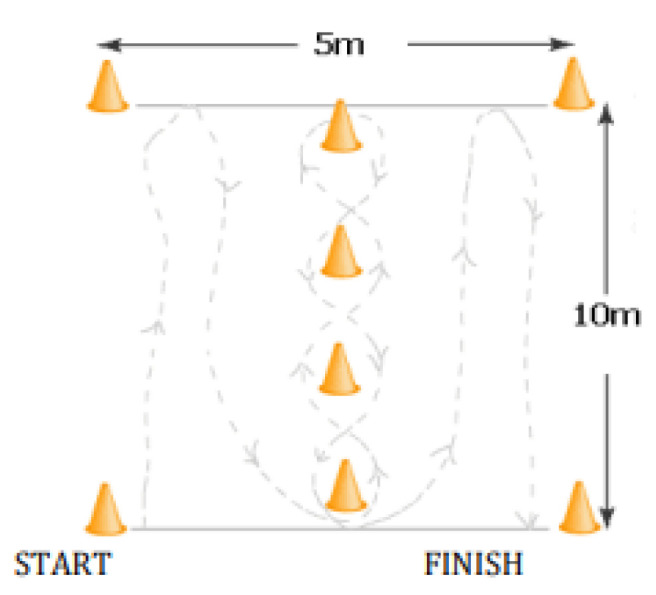
Illinois Agility Test.

**Table 1 healthcare-11-00261-t001:** Training Overview.

	Main Focus	Frequency Sessions Per Week	Sample Daily Program
Calisthenic Training	Conditioning and cardiovascular fitness	3 days per week 1.5 h session	Running, body weight exercises—Push-ups, air squats, flutter kicks, etc.
Functional Fitness Training	Multi-joint movements, high intensity, varied intensity changes	3–5 days per week 1.5 h session	Defensive tactics/arrest and control training focus—Striking, grappling, ground defense, multiple officer scenarios
Strength Training	Core/Basic resistance training	3 days per week 0.5 h session	Sessions followed mat room training (defensive tactics/arrest and control training), and recruits incorporated one of the four lifts (squat, deadlift, bench press, and overhead press) at the beginning of these sessions and then chose accessory work

**Table 2 healthcare-11-00261-t002:** Distribution of Gender by time.

Gender	2001 (%)	2011 (%)	2018 (%)
Female	14 (14.0%)	13 (13.7%)	15 (13.0%)
Male	86 (86.0%)	83 (86.3%)	100 (87.0%)

**Table 3 healthcare-11-00261-t003:** Summary statistics on performance measures.

	Calisthenics Mean (SD) 2001			Functional Fitness Mean (SD) 2011			Strength Training Mean (SD) 2018		
	Pre	Post	Δ%	Pre	Post	Δ%	Pre	Post	Δ%
Age (years)	31.7 (5.1)	31.8 (5.1)	--	31.3 (5.6)	31.5 (5.5)	--	32.5 (5.0)	32.7 (5.0)	--
Weight (kg)	81.6 (14.5)	79.5 (13.3)	−0.9 (3.0)	85.9 (14.2)	85.0 (13.4)	−0.5 (4.0)	84.9 (13.8)	85.9 (13.8)	1.5 (3.0)
Body fat percentage (%)	18.3 (6.1)	15.9 (6.0)	−12.4 (13.6)	17.1 (5.7)	13.2 (4.6)	−18.3 (19.2)	16.6 (6.0)	13.6 (5.0)	−14.6 (19.5)
Trunk flexibility (cm)	50.0 (8.3)	53.4 (7.4)	5.71 (3.83)	50.2 (7.1)	52.1 (6.9)	3.83 (7.28)	50.8 (8.2)	51.9 (8.7)	2.4 (6.7)
Bench (kg)	80.6 (22.0)	86.5 (25.1)	10.4 (11.1)	83.1 (23.7)	91.4 (23.2)	8.9 (9.1)	89.0 (23.6)	98.2 (24.8)	11.1 (9.7)
Bench relative to weight percentage (%)	99.0 (23.7)	108.8 (27.4)	11.4 (11.7)	97.3 (25.5)	108.5 (25.4)	10.2 (12.2)	105.2 (24.8)	114.8 (26.3)	10.0 (10.6)
Agility (sec)	17.0 (1.2)	16.7 (1.2)	−2.4 (3.7)	17.3 (1.3)	16.6 (1.2)	−2.9 (3.7)	16.9 (1.2)	16.4 (1.2)	−2.4 (3.0)
Cardiovascular endurance (min)	12.2 (2.1)	11.1 (1.8)	−7.1 (6.6)	12.4 (2.19)	11.4 (1.9)	−6.2 (5.6)	11.7 (1.6)	11.2 (1.4)	−3.6 (8.2)

*p*-Value < 0.05.

**Table 4 healthcare-11-00261-t004:** Effect of predictor variables on Performance Measures.

Dependent Variable	Weight	Flexibility	Bench Press	% of Bench Press Relative to Weight	Cardiovascular Endurance (Run)	Agility	Body Fat %
	β (SE)	β (SE)	β (SE)	β (SE)	β (SE)	β (SE)	β (SE)
Intercept	176.83 * (10.56)	20.30 * (1.08)	237.89 * (13.63)	134.90 * (7.24)	9.08 * (0.60)	14.49 * (0.31)	10.33 * (1.80)
Gender ^‡^—Female	−32.80 * (4.81)	2.34 * (0.49)	−98.73 * (6.21)	−40.89 * (3.30)	2.73 * (0.27)	2.33 * (0.14)	8.99 * (0.82)
Age	0.23 (0.32)	−0.03 (0.03)	−1.48 * (0.41)	−0.96 * (0.22)	0.08 * (0.02)	0.07 * (0.01)	0.21 * (0.05)
Time ^†^—2011	9.39 * (4.15)	0.10 (0.43)	4.57 (5.36)	−2.23 (2.84)	0.29 (0.24)	0.28 * (0.12)	−1.14 (0.71)
Time ^†^—2018	6.71 (3.97)	0.35 (0.41)	18.74 * (5.12)	6.61 * (2.72)	−0.56 * (0.23)	−0.18 (0.31)	−1.83 * (0.31)

^‡^ Ref. Category = “Male”, ^†^ Ref. cat = “2001”, * *p*-value < 0.05.

## Data Availability

Not applicable.

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
