# Peer review of "Evolution of Physical Training in Police Academies: Comparing Fitness Variables"

_healthcare, 2023, doi:10.3390/healthcare11020261_

Round 1
Reviewer 1 Report
This article is outlining great work and data analysis. Interesting statistics are pointed out and good conclusions on training programs are outlined. But some items would need to be revisited to further improve it:
1) While differentiating between groups (using a time and gender perspective) was relevant, it could also be relevant to add comments on the diversity within the chosen LEO groups since police officers are multidimensional human beings. More specifically, my question is: On top of age and gender, were the studied groups found to be similar (or not) in terms of other demographic profiles (e.g. racial or cultural background) in 2018 vs 2001 to 2011)? If it is not known because of the available information, the authors might comment (in the discussions) on the potential impact of disparity (or not) in other characteristics (in 4.1). It could also be mentioned in the limitation section (4.3) since it shouldn't be silent on diversity, but rather it should be if it was taken into account in the reflections and be presented as a potential follow-up or else. That would add to the context and make the methodology and its boundaries (or limitations) more robust. [For example, other studies have shown that female, Hispanic, or Black officers are making significantly fewer stops, arrests, and use of force than white male colleagues. While this is a more complex question, it has a direct impact on LEO occupational stressful tasks, highly physically and mentally demanding, for which the training as pointed out by the authors is justified.]
2) In the conclusions section, it could be useful to be more explicit on the fact that while each mode adds a positive impact, some were better (...more impactful) than others. For example, in the text, the authors concluded the discussion section (4.2) saying that "it is possible that recruits could benefit from an additional training stimulus or program adjustments to further develop aerobic capacity depending on documented physical job demands." It might be relevant to recall such a "conclusion" in the Conclusions.
3) The article could benefit from another language revision. Typos, dashed letters ("a"), double spaces (" __ "), and/or double dots ("..") are still present in the document. Correcting them would allow the reader to further appreciate the important work that was done.
Author Response
1) While differentiating between groups (using a time and gender perspective) was relevant, it could also be relevant to add comments on the diversity within the chosen LEO groups since police officers are multidimensional human beings. More specifically, my question is: On top of age and gender, were the studied groups found to be similar (or not) in terms of other demographic profiles (e.g. racial or cultural background) in 2018 vs 2001 to 2011)? If it is not known because of the available information, the authors might comment (in the discussions) on the potential impact of disparity (or not) in other characteristics (in 4.1). It could also be mentioned in the limitation section (4.3) since it shouldn't be silent on diversity, but rather it should be if it was taken into account in the reflections and be presented as a potential follow-up or else. That would add to the context and make the methodology and its boundaries (or limitations) more robust. [For example, other studies have shown that female, Hispanic, or Black officers are making significantly fewer stops, arrests, and use of force than white male colleagues. While this is a more complex question, it has a direct impact on LEO occupational stressful tasks, highly physically and mentally demanding, for which the training as pointed out by the authors is justified.]
No, racial and cultural background were not provided, this was added to the limitations.
2) In the conclusions section, it could be useful to be more explicit on the fact that while each mode adds a positive impact, some were better (...more impactful) than others. For example, in the text, the authors concluded the discussion section (4.2) saying that "it is possible that recruits could benefit from an additional training stimulus or program adjustments to further develop aerobic capacity depending on documented physical job demands." It might be relevant to recall such a "conclusion" in the Conclusions.
Expanded the conclusion to infuse the findings more
3) The article could benefit from another language revision. Typos, dashed letters ("a"), double spaces (" __ "), and/or double dots ("..") are still present in the document. Correcting them would allow the reader to further appreciate the important work that was done.
Reviewed in Grammarly for such errors

Reviewer 2 Report
The authors perform a study to evaluate the effectiveness of different training approaches in police academies. This area is shown to be relevant; it is increasingly important to understand the current trends in police officer fitness state. Overall, the paper is interesting to read. Study of relevant literature is performed. Paper structure makes it easy to follow.
However, there are some drawbacks that should be considered to improve the quality of the paper before publishing:
1) “Statistical Approach” section (Line 147) is very brief. Need to expand that section to include reasons why specific statistical approaches were chosen for the study including references. Why was SAS 9.4 software chosen for the statistical analysis?
2) Results section need to be split into paragraphs. Currently it contains a massive single paragraph (Lines 166 – 194) that makes it hard to read. Please identify key outcomes from results and group the numbers based on those distinguished results.
3) Performance Variables section (line 226). I assume that should be highlighted as a heading and numbered. Please revise.
4) Conclusion section is very brief. Please add some information on how this study can be expanded in the future. What are the directions for future research based on the outcomes of your study?
Once this is amended, the paper will be suitable for publication in this journal.
Author Response
1) “Statistical Approach” section (Line 147) is very brief. Need to expand that section to include reasons why specific statistical approaches were chosen for the study including references. Why was SAS 9.4 software chosen for the statistical analysis?
This has been expanded. Most up to date version of SAS was chosen due to ease of analysis for GLM and familiarity with the software.
2) Results section need to be split into paragraphs. Currently it contains a massive single paragraph (Lines 166 – 194) that makes it hard to read. Please identify key outcomes from results and group the numbers based on those distinguished results.
Results section now separates all variables of interest in individual paragraphs while guiding the reader to Table 4 for all pertinent data.
3) Performance Variables section (line 226). I assume that should be highlighted as a heading and numbered. Please revise.
This has been revised. Performance Variables are now listed as “4.3” in this section.
4) Conclusion section is very brief. Please add some information on how this study can be expanded in the future. What are the directions for future research based on the outcomes of your study?
Expanded on
